# Rhinovirus Inhibitors: Including a New Target, the Viral RNA

**DOI:** 10.3390/v13091784

**Published:** 2021-09-07

**Authors:** Antonio Real-Hohn, Dieter Blaas

**Affiliations:** Center for Medical Biochemistry, Vienna Biocenter, Max Perutz Laboratories, Medical University of Vienna, Dr. Bohr Gasse 9/3, A-1030 Vienna, Austria

**Keywords:** picornavirus, enterovirus, rhinovirus, viral RNA structure, pyridostatin, G-quadruplex, anti-viral, drug candidate, uncoating, inhibition

## Abstract

Rhinoviruses (RVs) are the main cause of recurrent infections with rather mild symptoms characteristic of the common cold. Nevertheless, RVs give rise to enormous numbers of absences from work and school and may become life-threatening in particular settings. Vaccination is jeopardised by the large number of serotypes eliciting only poorly cross-neutralising antibodies. Conversely, antivirals developed over the years failed FDA approval because of a low efficacy and/or side effects. RV species A, B, and C are now included in the fifteen species of the genus *Enteroviruses* based upon the high similarity of their genome sequences. As a result of their comparably low pathogenicity, RVs have become a handy model for other, more dangerous members of this genus, e.g., poliovirus and enterovirus 71. We provide a short overview of viral proteins that are considered potential drug targets and their corresponding drug candidates. We briefly mention more recently identified cellular enzymes whose inhibition impacts on RVs and comment novel approaches to interfere with infection via aggregation, virus trapping, or preventing viral access to the cell receptor. Finally, we devote a large part of this article to adding the viral RNA genome to the list of potential drug targets by dwelling on its structure, folding, and the still debated way of its exit from the capsid. Finally, we discuss the recent finding that G-quadruplex stabilising compounds impact on RNA egress possibly via obfuscating the unravelling of stable secondary structural elements.

## 1. Introduction

The common cold symptoms caused by rhinoviruses (RVs) differ very much from the severe symptoms associated with infections by some select representatives of other *Enterovirus* genera (e.g., poliovirus, enterovirus 71, enterovirus D68, etc.). Nevertheless, the high similarities of their genome sequences led the International Committee on Taxonomy of Viruses to include the species RV-A, RV-B, and RV-C in the genus *Enterovirus* (see e.g., ref. [1]). Where appropriate, we have thus taken the liberty to extrapolate findings from other members of this genus onto RVs without explicit statement. However, it should be kept in mind that one of RVs’ most distinct physicochemical properties is their inactivation at a slightly acidic pH, which was originally taken as a basis for classification [2]. This instability at a low pH even led to considering acid-buffered saline as anti-RV nose drops [3]. Furthermore, it is important to keep in mind that differences among the *Enterovirus* genera, RV species, and even individual RV types often result in disparities with respect to the inhibitory activity of a given drug candidate. In particular, the more recently discovered RV-Cs behave quite differently from RV-A and RV-B; for example, they do not possess pocket factors and, consequently, are not inhibited by several of the compounds mentioned in Table 1. This fact is enormously complicating the development of a ‘pan- *Enterovirus*’, or, at least, a ‘pan-rhinovirus’ inhibitor.

RV infections usually take a mild course with the pertinent signs of a respiratory tract irritation, typically a runny nose, a sore throat, and sometimes a slightly increased body temperature; these symptoms usually resolve within around a week (for a review on all aspects of rhinoviruses and rhinovirus infections, see ref. [4]). Nevertheless, the economic impact of RV infections is huge because of both sick leaves and spending for medication that only slightly alleviates the unpleasant symptoms. In 2003, the costs of respiratory viral infections, with approximately 50% of them caused by RVs, was estimated to be approximately $40 billion annually in the US [5]. As effective treatment options are still lacking, this number has probably not changed very much over the past years. Notably, RVs might become life-threatening for patients suffering from cystic fibrosis, asthma, chronic obstructive pulmonary disease, or other respiratory ailments [6].

Protection against RV via vaccination was contemplated early on and still is up to the present day [7,8]—for the original article, see ref. [9]. However, it is seriously hampered by the large number of RV types; at the time of writing, i.e., 1 September 2021, 169 distinct RV strains were listed (https://www.picornaviridae.com/sg3/enterovirus/enterovirus.htm). Attempts at immunising with mixtures of inactivated prototype strains [10] or with recombinant viral capsid proteins have shown to induce only weakly cross-neutralising antibodies in mice [11,12], and patient data are relatively scarce [13]—more recently reviewed in ref. [14]. Unfortunately, mouse models (e.g., see ref. [15]) have not proved very useful for assaying the inhibition of viral infection in vivo, since the symptoms of infection are quite limited and usually viral titres surpass the inoculum only marginally, if at all.

Early on, preventing or at least significantly shortening the duration of the symptoms of RV infection via inhibiting crucial steps of viral replication have been sought, and various drug candidates were developed that are directed toward attachment, entry, uncoating, RNA synthesis, proteolytic processing of the polyprotein, and viral assembly (Figure 1, Tables 1–3; for excellent in-depth reviews on *Enterovirus* inhibitors see refs. [16,17]). More recently, cellular targets have moved into focus and research along these lines is taking up pace [18,19,20,21]. The inhibition of factors necessary for viral reproduction provided by the infected cell might avoid the emergence of escape mutants, a pertinent problem with drugs blocking the function of a viral protein; however, evidently, side effects might be more severe. Furthermore, compounds encumbering access of the virus to the cell, such as iota-carrageenans [22], reducing the number of infectious units via aggregation with highly charged multivalent nanoparticles [23], or obstructing the viral receptor binding sites with dendrimers [24], are other directions of anti-viral research (Table 3). It is noteworthy that the above dendrimers not only block EV-A71 but also the unrelated human immunodeficiency virus (HIV), indicating that compounds with even broader antiviral specificity, including for RV, might be identified in the future. As briefly described below, these strategies have recently been expanded onto small molecules, forming filaments ‘catching’ the virus particles, similar to the otherwise completely unrelated neutrophil extracellular traps [25].

## 2. Targeting the Viral Capsid

Amongst the best-known RV inhibitors are multi-ring organic compounds, which, early on, were mainly isoxazole derivatives. These chemicals act via stabilising the icosahedral protein shell against the structural changes required for release of the viral genome. In some cases, they also impact on viral binding to intercellular adhesion molecule 1 (ICAM-1), the cognate receptor of the majority of RVs [26,27,28,29,30,31,32,33,34,35]. Due to their high affinity for a solvent-accessible hydrophobic pocket built from amino acid residues mostly contributed by VP1, the largest of the four capsid proteins, they displace the natural pocket factor, a fatty acid-like molecule found in most RVs [36]. Filling the void impedes the moving in of amino acid side chains, which is necessary for subsequent structural changes in the protein capsid, as are required for ordered uncoating. In vitro, heating to ≥50 °C in low-ionic strength buffers leads to RNA exit; when the experiment is carried out in the presence of such inhibitors, the ordered conformational change and release of the RNA is impaired, and the capsids become distorted [37]. The stabilising effect of such compounds has also been explored via molecular dynamics simulations and indicated a decrease in entropy [38]; counterintuitively, the binding of the drug was predicted to increase rather than decrease the compressibility [39]. More recent investigations found out how the bioavailable pyrazolopyrimidine OBR-5-340 binds to the pocket of pleconaril-resistant RV-B5 with a different geometry and specificity [40]; conversely, compound 17 fills another recently discovered solvent-accessible inter-protomer pocket in coxsackievirus B3 [41]. The latter similarly impedes the transition from the native virus into the A (i.e., ‘altered’ or ‘activated’) particle. Research along these lines is still active, and novel EV-A71 neutralising compounds, employing similar mechanisms, have recently been reported but not tested against RVs [42]. The feeble stabilisation of the virion by the natural pocket factors is useful to prevent premature uncoating. However, for infection, the pocket must be emptied at the right location and time to allow its collapse and RNA discharge [43,44,45]. This is also nicely seen for RV-A89, where the elution of the pocket factor with DMSO resulted in the side chain of M1222 (the first digit denotes the VP; here, VP1) to move into the now empty pocket (ref. [40] and to be published elsewhere). The above subtle conformational changes facilitate the conversion of the native virion into the A particle, manifesting in expansion, the thinning of the shell, and the opening of pores at the two-fold and three-fold symmetry axes, as reported for poliovirus [46], RVs, e.g., RV-A2 [45], and other *Enteroviruses* [43,46,47,48,49,50,51,52]. Of note, in RVs, the conversion of the native virion into the A particle is triggered by the acidic endosomal pH and/or the interaction with the receptor [51,53,54,55]. For some picorna-like insect viruses that lack pocket factors [56,57,58], as well as for echovirus 18 [59], in vitro triggering uncoating under specific conditions was shown to result in the loss of one or more pentamers [59,60]. This suggests that at least part of the RNA can egress in bulk through the resulting big orifice. On the other hand, heating RVs to ≥50 °C in low ionic strength buffers [61,62] was seen to result in empty but otherwise entire icosahedrons. Taken together, this suggests that one or more pentamers might move out and flip back again after RNA discharge, like an opening and closing lid. Dynamic capsid disassembly/reassembly has indeed been demonstrated for the insect pathogen Triatoma virus by using native mass spectrometry [63].

As shortly summarised above, a great number of capsid stabilisers, many based upon isoxazole [26], have been developed, with the best known being pleconaril (reviewed in ref. [64]). However, the clinical use of pleconaril has been limited to life-threatening infections with various *Enteroviruses* [65,66,67]; it was not approved as medication against the common cold because of its low effectiveness in reducing the duration of the disease and side effects, including counteracting hormonal birth control [68]. Given the usually mild pathogenicity of RVs, such drugs will be widely accepted only when completely side-effect-free. Notwithstanding, research on the above molecules furthered our understanding of the mechanistic of viral infection and of the intricate early interactions between RVs and the host cell. The three-dimensional structures of various virus/inhibitor complexes have been elucidated; examples are given in Table 1. 

## 3. Targeting Non-Structural Viral Proteins

Upon entering the cytosol of the infected cell, the viral ss(+)RNA is first translated into a ~230 kDa polyprotein that is co-translationally and autocatalytically processed by the viral proteinases 2A^pro^ and 3C(D)^pro^ into the structural and the non-structural proteins (reviewed in ref. [79]). The very first step in this processing cascade is the severing of P1, the precursor of the capsid proteins, from P2P3, the precursor of the non-structural proteins, by 2A^pro^. The subsequent cleavages are carried out by 3C^pro^ and its precursor 3CD^pro^. During RNA encapsidation, VP0 is processed into VP4 and VP2, probably under participation of the RNA [80] and low vesicular pH, at least in the case of poliovirus [81]. Obviously, the specificity of 2A^Pro^ and 3C(D)^Pro^ makes them ideal targets for inhibitors. Of note, several cellular proteins cleaved by the viral proteinases have been identified. For example, 2A^pro^ inactivates the translation initiation factor eIF4G, shutting off cap-dependent translation [82,83,84,85,86,87]. Other targets are cytokeratin 8 [88], the mitochondrial antiviral signalling (MAVS) protein (cleavage shown for EV-A71), and serum response factor (cleavage shown for CoxB3) [89]. Efforts toward blocking the activity of the above enzymes led to the identification of several inhibitors (Table 2).

Recently, high-throughput affinity selection mass spectrometry has been used to identify small molecules binding 3C(D)^Pro^ by screening 100,000 compounds [104]. Fifty per cent of the few compounds identified exhibited inhibitory activity. It remains to be seen how inhibitors revealed by this novel technology compare to those already available, such as rupintrivir (ref. [105] and reviewed in refs. [106,107]). It is of note that 2A^pro^, as well as 3C^pro^, are cysteine proteinases with a trypsin-like fold [108,109] and are, therefore, distinct from cellular proteinases. Out of the various known inhibitors, some examples are listed in Table 2. 

The RNA-dependent RNA-polymerase 3D^pol^ and the accessory 2C^ATPase^, involved in the synthesis of ss(−)RNA and, later in infection, of massive amounts of ss(+)RNA, have been targeted as well. They are part of the membrane-associated replication complex (reviewed in ref. [110,111]). For inhibiting 3D^pol^, nucleoside and nucleotide analogues have been developed. Since RNA-dependent RNA polymerases do not exist in the host cell, this viral enzyme is also ideally suited as a target. Nevertheless, a high specificity of the drug candidates is required to avoid dangerous off-target effects, e.g., incorporation into cellular nucleic acids. On the other hand, a great number of RNA viruses possess such polymerases, and several of them might be affected by the same compounds [112]. Examples are given in Table 2.

As discussed in more detail below, the helicase 2C^ATPase^ appears to be involved in threading the newly synthesised RNA into partially assembled virions. It has been known for a long time that guanidine hydrochloride inhibits the replication of several picornaviruses via acting on 2C^ATPase^, but its toxic nature has precluded its use as a drug [113]. However, 2C^ATPase^ is also inhibited by the less toxic (S)-fluoxetine and derivatives [114], but not by (R)-fluoxetine, indicating stereospecificity. The binding site of (S)-fluoxetine has been modelled, but the mechanism of inhibition is not yet clear [95]. Dibucaine and its derivatives inhibit the ATPase activity of the EV-A71 2C^ATPase^ and, consequently, the RNA unwinding. Again, the binding site was modelled [115], but so far was not verified experimentally. 

In addition to the inhibitors of infection acting on the virus by mechanical aggregation, entrapment, receptor blockage, and on endosomal uptake and acidification, several inhibitors of cellular proteins are listed in Table 3. Noteworthy are those that act on functions necessary for viral replication. For example, the cellular thiol hydrolase PLA2G16 was identified in genetic screens via impairment of the viral infection. PLA2G16 is important for viral RNA transfer from endosomes into the cytosol [18,19,116]. It appears to prevent the premature detection of viral damage of the endosomal membrane and thus autophagic elimination of endosomes containing uncoating virus. LEI110 blocks PLA2G16 activity, but it is not reported whether it inhibits viral infection [117]. Presumably, drug candidates currently being developed by the pharmaceutical industry are based upon this lead compound. Of note, the knockout of PLA2G16 protected against virus infection [18] but also prevented obesity in mice [118], demonstrating how complex off-target effects may be.

The heat shock protein HSP90 was shown to be required for the correct folding of viral proteins, and its inhibition reduced the viral yield of several *Enteroviruses* [119,120,121], as also shown for RVs [122]. Itraconazole and enviroxime act on cellular enzymes involved in the synthesis of the membranous material required for assembly of the replication complex [123]. VP0, and consequently VP4, becomes myristoylated by cellular N-myristoyltransferases. This modification is required for viral assembly and capsid stability. Consequently, interfering with myristoylation sensibly reduces viral production. DDD85646 inhibits the cellular enzymes and has a similar effect as knocking down the two isoenzymes [124,125]. 

Contemplating the various drug targets and the corresponding drug candidates in Figure 1 and Table 1, Table 2 and Table 3, it becomes clear that one, namely the viral RNA genome that has attracted attention as a point of attack in the case of influenza A virus, Zika virus, and herpes simplex virus 1 [126,127,128], has so far not been considered in the case of picornaviruses [94,129,130]. In the following, we shall thus discuss the possibility of impacting on the stability of specific secondary structural elements of the viral RNA. We shall also briefly present a related unexpected discovery of virus trapping by filaments formed by the small molecule Pyridostatin (PDS) under specific conditions (ref. [25], Table 3).

## 4. The Rhinoviral RNA 

The RV ss(+)RNA genome is approximately 7,300 nucleotides in length. At its 5′-end, it carries a peptide (VPg) of some 23 amino acid residues covalently attached via a tyrosine phosphodiester linkage. At its 3′-end is a genome-encoded poly-(A) tail of between 50 and 200 adenosines [158]. From this genomic ss(+)RNA, ss(−)RNA is synthesised in a membrane-associated replication complex. Translation of the ss(+)RNA results in a polyprotein that is autocatalytically cleaved by the viral proteinases 2A^pro^, 3C^pro^, and its precursor 3CD^pro^ into four structural and seven non-structural proteins (not counting their precursors). 

## 5. RNA Encapsidation

Complementation experiments with different coxsackievirus strains point to a role of the helicase-like 2C^ATPase^ protein, a component of the replication complex, in encapsidation. Protein 2C^ATPase^ interacts directly with VP3, suggesting a possible involvement in threading the RNA into the assembling capsid [159,160]. Unexpectedly, 2C^ATPase^ was also reported to play a role in uncoating [161] and host range [162]. The former might be taken to indicate that it impacts on how the RNA is encapsidated and consequently released from the viral shell during infection. Ref. [163] summarises the current knowledge on 2C^ATPase^ structure and function. Filling the capsid is not considered as energy-driven RNA pumping [164]. Encapsidation might be initiated by VPg and 5′-proximal RNA sequences binding to pentamers; indeed, replicating poliovirus ss(+)RNA could be UV-crosslinked to pentamers, and Pfister and colleagues saw ‘capsid-like structures’ upon a negative stain electron microscopy of poliovirus replication complexes [165]. This supports the idea that the RNA is being delivered to the assembling protein shell during its synthesis but leaves us with the question of how natural empty capsids, also called ‘natural top component’ (NTC) in RVs based upon their sedimentation properties in sucrose density gradient centrifugation [61,62], are being assembled without the assistance of the RNA. Nevertheless, in vitro self-assembly was demonstrated for several picornaviruses, including bovine enterovirus [166], and was nicely depicted in a movie by using plastic models of pentamers equipped with magnets [167]. 

Early on, it was demonstrated that viral synthesis was occurring co-translationally and co-transcriptionally, and the in vitro assembly of infectious virions from the separate components, i.e., the viral capsid proteins and the RNA, was not achieved. For poliovirus, reproduction from viral RNA in vitro was accomplished, but required the presence of cellular membranous material, apparently for creating an assembly platform for the replication complex [168]. As pointed out above, the RNA likely starts interacting with building blocks of the capsid, which are probably protomers, or, even more likely, pentamers, as soon as its 5′-end decorated with VPg emerges from the replication complex. According to this view, the RNA starts folding and, by binding to protomers, it stabilises their interactions. This aids the assembly of more pentamers into a growing shell. Nevertheless, NTC apparently assembles without the aid of RNA, and, in the case of poliovirus, virus-like particles lacking RNA have been obtained even in a plant-based expression system [169].

Highly degenerate repetitive sequences in the RNA have been first identified by using the systematic evolution of ligands by exponential enrichment (SELEX) in the more distantly related parechoviruses to most probably constitute multiple packaging signals [170]; this was then also shown to apply to members of the genus *Enteroviruses* [171,172,173,174]. Indeed, close contacts between RNA and the protein capsid are seen in the 3D structures of several *Enteroviruses*, and their density corresponding to nucleotides could be clearly recognised (e.g., see ref. [171]). Nevertheless, because of the asymmetry of the RNA, the contacts must have a slightly different geometry in each case. Furthermore, RNA within individual virions apparently does not follow the same path. Rather, its passage from a given protomer to a defined symmetry-related protomer might differ. This means that RNA–protein interactions within a single protomer appear to be independent from their symmetry relation with the other protomers [175].

## 6. RNA Uncoating

Results of in vitro experiments on RNA release from virions suggest that the unstructured poly-(A) tail, most probably the last part of the RNA to become encapsidated, is the first to emerge from the protein shell [176,177]. Nevertheless, it is unlikely that the extensively structured rest of the RNA molecule, with its cloverleaf, the IRES element, and many other double-stranded regions [178], including very stable G-quadruplexes (GQs—see below), slips through one of the symmetry-related holes in the A particle shell or even through proteinaceous ‘umbilical connectors’, as suggested for poliovirus [179]. This would at least require complete unfolding, and it is not clear where the necessary energy would come from. At least part of it might stem from interactions with the receptor (ICAM-1) at a low pH, as observed for RV-B14; these lead to a rearrangement of the RNA around the two-fold axis, involving the loss of interaction with TRP 2038 [180]; this is in accordance with nanoDSF measurements of TRP fluorescence changes during the in vitro uncoating of RV-A2 via heating [181]. 

We believe it rather likely that the poly-(A) tail initiates exit, maybe via passing through one of the above-mentioned holes, with the bulk of the RNA then being released through a larger orifice created by the (reversible) swinging open of one or more pentamers; as mentioned above, particles lacking one or more pentamers were seen in cryo-EM images reconstructed without imposing icosahedral symmetry [59,176].

The RNA inside the viral shell is compact but not pressurised like dsDNA in some bacteriophages [182]. Nevertheless, it was suggested that viral RNAs are ‘unusually compact’ in order to fit into the limited space of the shell [183], but calculations show that they are definitely not under pressure [184]. As discussed below, alkali ions (Na^+^ versus K^+^) might modify this ‘compactness’.

The RNA genome maintains its overall globular shape even after the proteolytic removal of the capsid with proteinase K under mild conditions, as visualised with atomic force microscopy (AFM) in the case of poliovirus [185] and with rotary shadowing electron microscopy for RV-A2 (Figure 2a). In the above AFM experiments, increasing the temperature resulted in the partial unfolding of the spheroid RNA cores. Strikingly, some refolding occurred upon cooling, suggesting that at least small parts of the spherical outline are determined by secondary structural elements and not exclusively by the geometry of the capsid void. Unfolding by heating to 65 °C, for poliovirus [185], or to 90 °C, for RV-A2, followed by refolding, failed to re-create anything similar to the original core (Figure 2b). As pointed out above, this suggests that the nucleic acid most likely folds during its emergence from the replication complex, and that its progressive encapsidation involves intramolecular and intermolecular interactions with the protein.

## 7. Access of Intercalating Compounds Is Limited by the Compactness of the RNA Core

RNA-binding compounds have been present in the virologists’ toolbox for a long time: (i) acridine orange and neutral red are well known from early studies on the uncoating of several *Enteroviruses* [186,187], (ii) SYTO 82 and RiboGreen were mainly used to follow RNA release from virions in live-cell microscopy and capillary electrophoresis [188,189], and (iii) psoralen analogues were employed in RNA interactome mapping in the Zika virus, SARS-CoV-2, enterovirus D68 [190,191,192], and in RV-A2 for covalently cross-linking double-stranded regions in the RNA [176]. RNA was also labelled inside RV-B14 with N-acetyl-aziridine [193]. Delivery to the viral genome was either achieved via diffusion through the proteinaceous capsid wall during extended coincubation at 34 °C to 37 °C, where breathing, i.e., the dynamic opening of the capsid, is increased (e.g., in RV-A2, [189]), or via in vivo incorporation during viral replication (e.g., in poliovirus, ref. [188]).

As pointed out above, the diffusion of small molecular mass RNA double-strand-binding dyes to the viral RNA is obviously limited by the protein capsid. However, it is also limited by the RNA itself when condensed in a spherical core. As seen in Figure 3, with refolded RNA (red line), the fluorescence signal of the RNA-binding dye SYTO 82 decreases with an increasing temperature, reflecting the inverse correlation of dye-binding and temperature [181]. In isolated RNA cores (blue line), a similar signal decrease occurs, but a small fluorescence peak is seen at approximately 57 °C, suggesting that the dye intercalates into double-stranded regions that are apparently not accessible at a lower temperature. At a concentration of roughly 2 kg/L, the tightly folded RNA has a crystalline-like consistency [194], which obviously affects the diffusive penetration of small molecules into the RNA spheres. This idea is supported by the appearance of decapsidated but otherwise native viral RNA (Figure 2a) and of denatured and refolded RNA (Figure 2b). As indicated above, the latter shows a much stronger staining with SYTO 82 (Figure 3—red curve). Since there is no reason to believe that it possesses more double-stranded regions available for dye intercalation, the same regions must rather become more easily accessible because of some general expansion. The temperature dependence of the staining suggests that (i) at a low temperature, the compactness of the RNA impedes access of the dyes, (ii) increasing the temperature increases the accessibility of double-stranded regions, and (iii) further increasing the temperature reduces the signal because the affinity of the dye for the RNA generally diminishes and double-stranded segments unfold [195].

## 8. Targeting the Viral RNA

As pointed out above, the viral RNA’s folding is most likely modulated by its interactions with the inner surface of the protein shell [196], although its overall shape persists even in the absence of the protein capsid (Figure 2a). Furthermore, folding must inevitably occur in sequential order, i.e., starting from the 5′-end emerging from the replication complex. Its appearance utterly differs when denatured and refolded (Figure 2b). Nevertheless, it cannot be excluded that the spherical shape is stabilised by peptides remaining bound even after extensive proteolysis. It has been demonstrated that seemingly deproteinated ribosomes still contain residual peptides refractive to removal [197]. At the inner face of the protein capsid, the RNA is intricately interwoven with amino acid residues of VP4 and the N-terminal extensions of the other VPs [45]. Due to its lack of symmetry, RNA sequences contacting the symmetry-related identical protein patches of the protomers appear to impinge various conformations onto the latter, which translates into disorder in symmetrically averaged cryo-EM reconstructions. However, as pointed out above, such differences can be identified by the 3D classification of single protomers computationally excised from a cryo-EM map [175].

So far, we have only discussed double-stranded RNA stretches interrupted by single-stranded regions, usually appearing as loops in folding predictions [178]. In addition to these structural elements, picornaviruses, among other viruses, contain well defined internal ribosome entry sites (IRES), the clover leave [198,199] with characteristic secondary structures important for transcription/translation, the above-mentioned poly-(A) tail, and a number of sequence stretches with the propensity to form G-quadruplexes [200]. Although the importance of these non-canonical folds of G-rich domains in viruses is unclear, they are being increasingly considered as therapeutic targets, e.g., in the context of the COVID-19 pandemic-causing coronavirus-2 [201], reviewed in ref. [202]. Indeed, these intrinsically stable folds can be further stabilised by chemicals, including PDS, PhenDC3 (Figure 4), and other structurally related and unrelated compounds (reviewed in ref. [203]). Such GQ stabilisers are currently under focus for their potential use as anticancer drugs. For a recent review, see ref. [204].

## 9. Stabilising GQs in Viral RNA

Harutyunyan and colleagues assumed that RNA exit from the shell requires at least some unfolding [176]. They thus attempted, in RV-A2, to encumber RNA egress via introducing double-strand cross-links with 8-methoxypsoralen and UV irradiation before triggering in vitro uncoating via heating to 56 °C. They reasoned that preventing unravelling double-stranded regions would slow or prevent complete RNA exit altogether. Experiments along these lines indeed suggested that the 3′-poly-(A) tail became exposed to nucleases prior to more upstream regions, and that residual RNA with a rod-like appearance remained trapped inside the virions. The early exposure of 3′-regions of the RV-A2 RNA during the release process was also found in vivo [177]. Of note, the late Rossmann laboratory observed enterovirus D68 particles containing similar rod-like structures. However, they interpreted them as incompletely assembled virions [43]. 

Based upon the above crosslinking results, one might assume that the stabilisation of specific secondary structure elements of the RNA might similarly hamper its exit from the viral shell. As mentioned above, typical nucleic acid sequences that can be specifically targeted are GQs (recently reviewed in refs. [204,205]). In silico predictions indeed indicated the presence of such sequences in RVs, as in many human viruses refs. [200,202,206,207] and unreviewed manuscript preprint [157]; their use as targets in antiviral therapy is increasingly being contemplated [127,201], but has so far not been sought for RVs.

Upon co-incubation in a Tris-HCl buffer, the small molecule GQ binder PDS (Figure 4) indeed showed inhibition of RV-A2 infection. However, control experiments revealed that this was due to the PDS forming long fibrils trapping the virus in a net-like structure (Figure 5a from ref. [25] with permission). Although entirely different with respect to the filaments’ composition, the ‘catching’ mechanism resembles that of neutrophil extracellular traps; recently reviewed in ref. [208]. Apparently, the particular arrangement of the positive charges in the tris(hydroxymethyl) aminomethane (Tris) favours the stacking of PDS molecules to form a filamentous network; this effect is absent when the Tris-HCl buffer is replaced with phosphate-buffered saline (PBS), water, or Dulbecco’s modified Eagle’s medium (DMEM); see Figure 5b taken from ref. [25] with permission. As discussed below, the avoidance of Tris allowed the investigating of a possible effect of PDS on RNA exit in the complete absence of filament formation.

## 10. Inhibiting RNA Exit from Virions with PDS

The Eckard Wimmer lab showed that cell transfection with RNA containing a massively recoded poliovirus P1 sequence (i.e., encoding the capsid proteins) produces a variant with an infectivity comparable to that of the native virus, provided that the encoded amino acid sequence and the GC content were maintained [209,210]. This sequence alteration might, conceivably, lead to eliminating some GQs [200], as well as some of the ‘cryptic, sequence-degenerate, dispersed RNA packaging signals’ identified in EV-E [171]. At first sight, this might argue against an essential role for these sequence stretches in the virus replication cycle. Nevertheless, at least in the case of PV, recoding the w.t. P1 sequence into the sequence EU095952 (P1-Max), as published by Song et al. [209], created new GQs, as predicted by QGRS mapper [211]. Their number, but not their positions, were preserved (Figure 6). This might similarly apply to the cryptic packaging signals [171]. 

The mere presence of sequences predicted to adopt a GQ conformation does not attest to their corresponding folding when packed inside the capsid or when located in the cytoplasm. Therefore, it is fundamental to experimentally assess whether the predicted GQs are definitely adopting a genuine GQ fold.

From the 11 sequence stretches with a predicted propensity to form GQs in RV-A2 [211], two were selected for analysis: the one with the highest score (G20; position 1038–1064) and the one with the lowest score (G11; position 2048–2074). Both are bi-G-tetrad GQs [212]: G20 is a canonical GQ, and G11 is a non-canonical zero-nucleotide loop GQ [213]. The adoption of GQ folds was experimentally confirmed for oligonucleotides derived from the above sequences by NMR analysis, hydrogen–deuterium exchange, circular dichroism, and fluorescence displacement assays [unreviewed manuscript [157]].

The effect of PDS on purified RV-A2 was then studied in PBS after co-incubation at 34 °C and 4 °C, respectively. The higher temperature strongly enhances capsid breathing, which allows low molecular mass molecules such as RiboGreen, N-acetyl-aziridine, and 8-methoxypsoralen to diffuse into the virion and interact with the viral genome [176,193,214]; the molecular diffusion of such small molecules through the capsid wall is very low at 4 °C. RV-A2 was thus incubated with PDS at these respective temperatures, and accessibility of the RNA for SYTO 82 was determined [181]. The RNA genome of the virions loaded with PDS via incubation at 34 °C already became accessible at a lower temperature, but released the RNA at the same temperature as the sample that had been incubated with PDS at 4 °C. This might reflect a modification of the interactions between the capsid protein and the RNA by bound PDS molecules.

Work by Harutyunyan et al. [176] showed that the in vitro and in vivo (i.e., in infected cells) RNA exit from RV-A2 was impaired by cross-linking double-stranded regions of the viral genome via incubation with the capsid-permeable 8-methoxypsoralen followed by UV cross-linking. This implies that PDS might act similarly by non-covalently stabilising at least some of the GQs within the viral RNA. In vitro, the RNA exited at the same temperature in the samples treated with PDS at 34 °C and at 4 °C. However, in vivo, RNA egress from the ‘34 °C sample’ was impaired; indeed, the pre-incubation of RV-A2 with PDS at 34 °C followed by challenging HeLa cells revealed a low, dose-dependent inhibition of infection, duplicated by another GQ-binding compound, PhenDC3 (Figure 4) and (unreviewed manuscript [157]).

Above, the effect of PDS on the genomic RNA was only studied in PBS that, in addition to 147 mM Na^+^, contains 3 mM K^+^. It is well-established that the nature of cations impacts on the stability of GQs and the folding of the RNA in general [215,216]. Indeed, free spherical RNA cores (as in Figure 2a) were seen when the capsid digestion and all other incubations were carried out in potassium phosphate buffer. Conversely, in sodium phosphate buffer, the RNA cores looked somewhat flattened and collapsed onto the mica surface (unreviewed manuscript [157]). When the same experiment was carried out with PDS added after capsid digestion in the potassium phosphate buffer, the appearance of the RNA core was unaffected and identical to that in Figure 2a; however, most strikingly, in the sodium phosphate buffer, the presence of PDS resulted in the RNA core taking on a rod-like shape almost 10 times as long as the viral diameter and with a lateral extension of roughly 5 to 10 nm (unreviewed manuscript [157]).

Recently, short- and long-range interactions were demonstrated for the genomic RNA of enterovirus D68 [192] and other RNA viruses [191,217]. Such long-range interactions are probably involved in maintaining the spherical outline of the deproteinated viral RNA. We tentatively interpret its above reshaping by PDS in the sodium phosphate buffer as the consequence of the interference with these interactions by a currently unknown mechanism. 

The rod-like appearance of the RNA in the presence of PDS was exclusively seen in the sodium phosphate buffer. As mentioned above, viral infectivity was modulated by the incubation of purified RV-A2 with PDS under conditions allowing for diffusion through the capsid wall (unreviewed manuscript [157]). In agreement with the strikingly different effect of sodium and potassium ions at the RNA level, a significant drop in infectivity was observed uniquely when the incubation with PDS was carried out in a sodium phosphate buffer (unreviewed manuscript [157]). We speculate that this might be due to the increased accessibility of the GQs present in the RV RNA. This view is supported by significantly more thioflavin T binding to deproteinated spherical RNA cores in sodium phosphate buffer, as compared to the potassium phosphate buffer (unreviewed manuscript [157]). Furthermore, roughly 50 times more PDS was found to associate with the viral RNA within the intact virus upon incubation in the sodium phosphate buffer than in the potassium phosphate buffer (unreviewed manuscript [157]). This might be taken to indicate that the ionic environment has a very high impact on the accessibility of GQs within the RV genome and the consequences on RNA egress.

## 11. Outlook

In the first part of this article, we provide a short and therefore necessarily incomplete review on potential viral and cellular drug targets and their corresponding inhibitors (Figure 1). This includes the well-known hydrophobic pocket within the protein shell [218] and another recently discovered druggable interprotomer pocket [41]. With respect to the non-structural viral proteins, well-defined targets are the 2C^ATPase^ [95], the 2A^pro^ [92], the 3C^pro^ [99], and the 3D^pol^ [103]. Virus binding can be specifically blocked with soluble receptor derivatives [149,150], and, more specifically, with highly charged multivalent nanoparticles [23,219], the polysaccharides iota-carrageenan [22,220], and PDS fibrils [25], either aggregating the virus or blocking its access to the cell. Additionally, virus uncoating can be prevented by the inhibition of membrane and cortical cytoskeleton rearrangements [148], activating the autophagy pathway [147], interfering with the maturation of endosomal vesicles [145], and, more specifically, for RVs only, by increasing the endosomal pH [221].

Recently, research has focused on the therapeutic potential of impacting GQ folds in DNA in cancer, particularly directed towards telomers [222]; for a general review, see e.g., ref. [223]. This recent popularity of GQs (an online search for ‘g-quadruplex’ returns more than 1.8 million hits) drove the bioinformatics search and discovery of GQs in the genomes of many viruses [200] and opened a new avenue towards targeting such non-canonical folds with suitable small synthetic molecules that impact on the stability of these structures. Resolving RNA secondary structures is necessary for transcription, translation, and genome exit from the viral shell. In addition to GQs, G-triplexes and G-hairpins that are believed to merely be intermediates of GQ formation were more recently confirmed as stable structures, although they are so far only well established in DNA molecules [224]. Of note, not only GQs but also G-triplexes can be targeted by PDS and PhenDC3 [225], extending the number of binding partners. The diffusion of PDS and PhenDC3 into the virion to interact with the nucleic acid and thereby affect the infectivity, now emerges as proof for the existence and the principal accessibility of such RNA structures in encapsidated RNA. Of note, to our knowledge, such compounds were only shown to impact on the transcription and/or translation of the targeted nucleic acids and not on RNA exit from a virus. The infection-inhibitory effect of GQ-binding compounds should be considered a way toward viral inactivation via impacting on the genome when still being inside the capsid. It is noteworthy that this likely preserves the capsid epitopes, which might be useful for stabilising vaccines. Nevertheless, it must be kept in mind that the many GQs present in cellular nucleic acids also constitute targets for such GQ-binding drug candidates with a high potential for side effects. Although PDS and PhenDC3 only weakly inhibited RV infection (unreviewed manuscript [157]), small synthetic compounds selectively binding secondary structure elements might become novel leads and motivate research in this direction. It might be necessary to increase their specificity for virus-specific sequences. The effective treatment of human immunodeficiency virus (HIV) and hepatitis C virus (HCV) has become possible with blends of drugs targeting different viral proteins [226]. We thus believe that of the many existing RV inhibitors with different viral and cellular targets, combinations should be tested for a more effective inhibition of RV infection. 

## Figures and Tables

**Figure 1 viruses-13-01784-f001:**
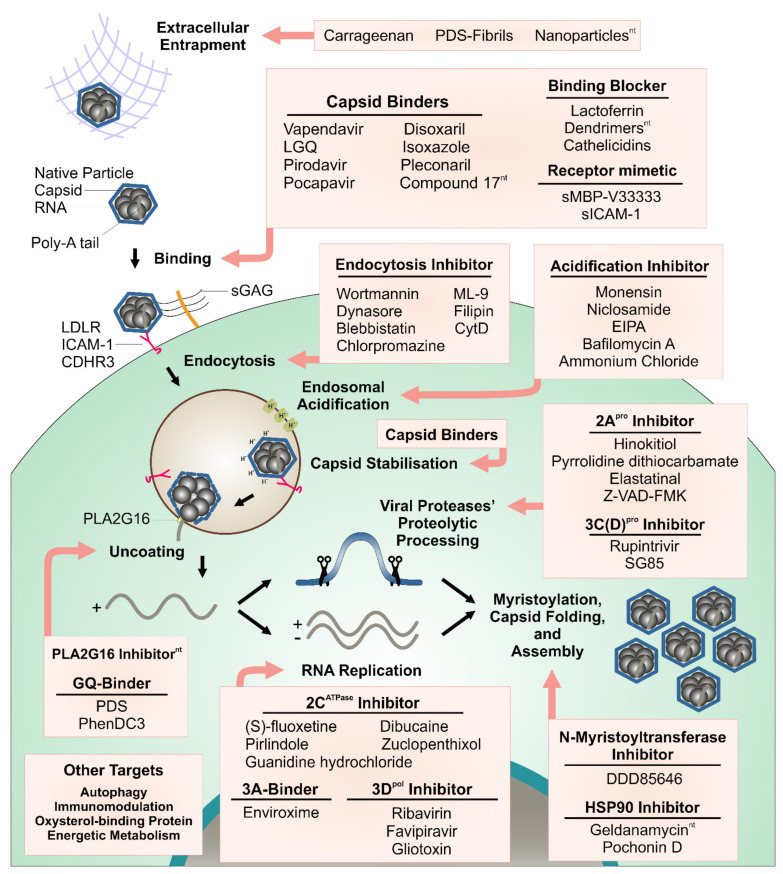
Anti-RV targets and corresponding inhibitors. Steps of the rhinovirus replication cycle targeted with antiviral compounds. Note that the publication on the pla2g16 inhibitor LEI110 does not mention anything related to RV (see text and Table 3). ^nt^, inhibition of RVs not explicitly demonstrated but tentatively inferred from inhibition of other *Enteroviruses*.

**Figure 2 viruses-13-01784-f002:**
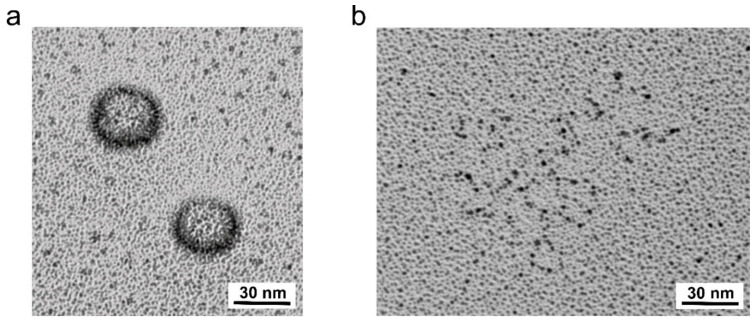
RV-A2 RNA morphology revealed by rotary shadowing electron microscopy. Purified RV-A2 in 100 mM potassium phosphate buffer (pH 7.4) was incubated with proteinase K for 12 h at 4 °C and divided into two aliquots. One was kept at 4 °C (**a**); one was kept for 10 min at 90 °C followed by slow cooling (**b**). The samples were mixed 1:1 with 200 mM ammonium acetate, 60% (*v*/*v*) glycerol, sprayed onto freshly cleaved mica chips, and coated with 0.6 nm platinum at an angle of 7° under high vacuum. Replicas were floated onto carbon-coated grids and imaged under an FEI Morgagni transmission electron microscope.

**Figure 3 viruses-13-01784-f003:**
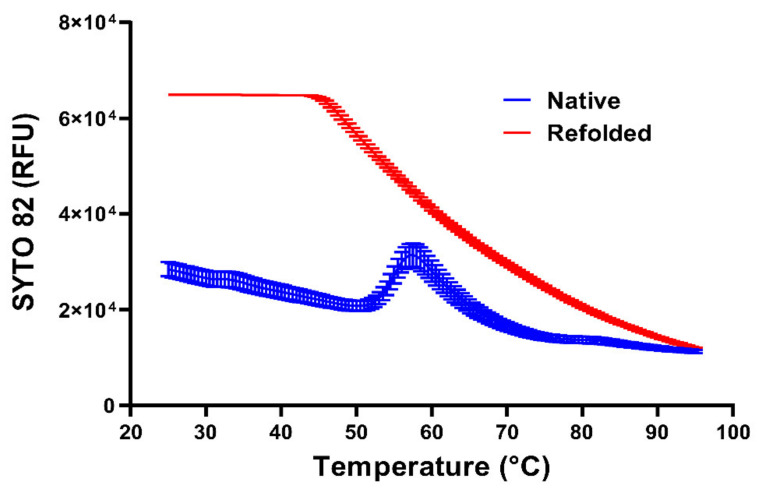
Temperature dependence of the accessibility of RV-A2 RNA for the intercalating dye SYTO 82. Purified RV-A2 in 100 mM potassium phosphate buffer (pH 7.4) was incubated with proteinase K for 12 h at 4 °C, as in Figure 2a. Samples were either kept at 4 °C (native—blue) or kept for 10 min at 90 °C followed by slow cooling (refolded—red) and mixed with SYTO 82 in 100 mM potassium phosphate buffer. Three aliquots of each experimental condition were disposed into the wells of a thin-walled PCR plate, the temperature was ramped from 25–95 °C at 1.5 °C/min, and the SYTO 82 fluorescence signal was recorded.

**Figure 4 viruses-13-01784-f004:**
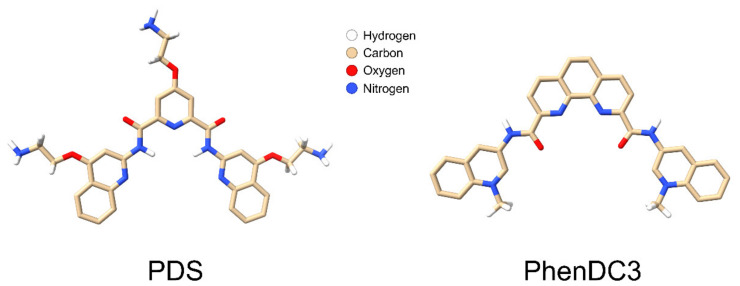
Representation of two GQ stabilisers. The structural formulas of PDS and PhenDC3 were prepared using Avogrado 1.2.0 software; atom colour codes are depicted. Hydrogens of the rings were omitted.

**Figure 5 viruses-13-01784-f005:**
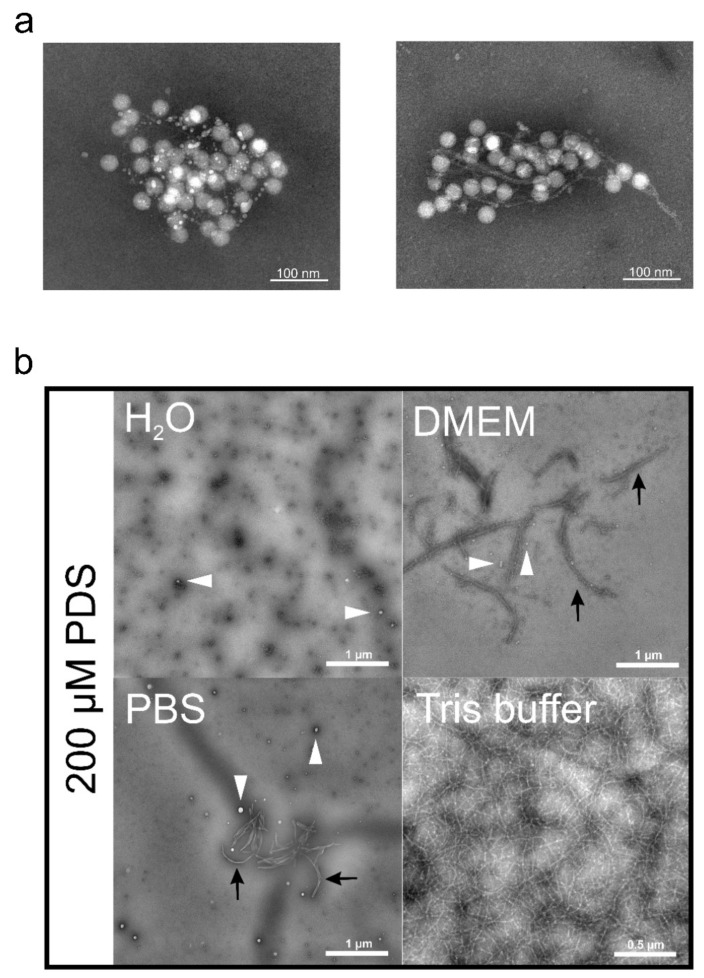
PDS Fibrils. (**a**) Two examples of RV entrapped by PDS fibrils. RV-A2 was mixed with PDS at 20 µM in Tris buffer, transferred onto EM grids, stained with 2% phosphotungstic acid (pH 7.4), and viewed with an electron microscope. Note the pearls-on-a-string-like appearance of the filaments in the left panel; (**b**) solvent effect on PDS fibril formation. PDS (200 µM final concentration) dissolved in the solutions indicated was applied onto EM grids, stained, and viewed as in (**a**). The white arrowheads point to larger amorphous aggregates; the black arrows point to ‘protofibrils’. Both panels are reproduced from ref. [25] with permission.

**Figure 6 viruses-13-01784-f006:**
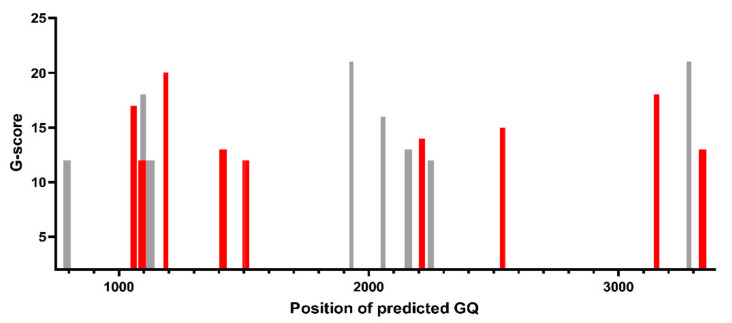
GQRS mapper prediction of GQs in the wt poliovirus 1 P1 region (NC_002058.3; grey) vs. the recoded sequence (P1-Max = EU095952; red). The P1 region comprises nucleotides from 746–3385 of the poliovirus polyprotein.

**Table 1 viruses-13-01784-t001:** List of compounds targeting a capsid pocket *.

Compound	PDB (Virus–Inhibitor Complex)	Clinical Trials Register	Reference
BTA798 (Vapendavir)	3VDD (RVA2)	NCT03024177 (w); NCT02367313 (nrp); 2014-001785-95 (cwr)	[69]
OBR-5-340 (LGQ)	6SK5 (RVB5)	-	[40]
R77975 (Pirodavir)	1PO2 (PV1); 1VBC (PV3)	-	[70,71,72]
SCH48973 (Pocapavir)	1EAH (PV2L)	-	[73]
Win-51711 (Disoxaril)	1D4M (CVA9); 1PIV (PV3); 3ZFE (EV71)	-	[35,74,75]
WIN-61209 (Isoxazole)	1QJU (RVA16)	-	[76]
Win-63843 (Pleconaril)	1C8M (RVA16); 1NCQ (RVB14), 1NCR (RVA16); 4WM7 (EVD68)	NCT00394914 (cwr)	[67,77,78]
Compound 17	6GZV (CVB3)	-	[41]

Abbreviations: nrp—no result posted; cwr—completed with result; w—withdrawn. Clinical trial repositories: ClinicalTrial.gov, accessed on 1 September 2021 and ClinicalTrialsRegister.eu, accessed on 1 September 2021. The virus mentioned denotes the virus–inhibitor complex structure present in the PDB register. * Note that all inhibitors locate to the hydrophobic pocket delimited mostly by VP1, except from compound 17, which binds to a recently identified inter-protomer pocket.

**Table 2 viruses-13-01784-t002:** List of compounds targeting viral non-structural proteins.

Target/Compound	Reference
2A^pro^	
Hinokitiol	[90]
Pyrrolidine dithiocarbamate	[91]
Elastatinal	[92]
Z-VAD-FMK	[93]
2C^ATPase^	
(S)-fluoxetine	[94]
Dibucaine	[95]
Pirlindole	[95]
Zuclopenthixol	[95]
Guanidine hydrochloride	[17]
3A	
Enviroxime	[96]
3C(D)^Pro^	
Rupintrivir	[97,98]
SG85	[99]
3D^Pol^	
Ribavirin	[100,101]
Favipiravir (T-705)	[102]
Gliotoxin	[103]

**Table 3 viruses-13-01784-t003:** List of compounds targeting non-viral proteins and proposed mechanism of action.

Compound	Proposed Mechanism	Clinical Trials Register	Reference
Lactoferrin	Sulfated glucosaminoglycan (sGAG) blocker	NCT01677702 (nrp); NCT01092039 (nrp)	[131,132,133,134,135]
Nanoparticles	Heparan sulfate proteoglycans mimetic Virion aggregation	-	[23]
PDS fibrils	Virion entrapment		[25]
Carrageenans	Virion entrapment	NCT01944631 (cwr); NCT04533906 (nrp)	[136,137]
LEI110 (*)	PLA2G16 inhibitor	-	[117]
Geldanamycin	HSP90 inhibitor	-	[119,138]
Pochonin D	HSP90 inhibitor	-	[122]
Wortmannin	PI 3-kinase inhibition	-	[139]
Dynasore	Dynamin GTPase inhibitor	-	[140]
Blebbistatin	Myosin ATPase activity inhibitor	-	[140]
Chlorpromazine	Prevent the assembly/disassembly of clathrin lattices	-	[140]
Filipin	Cholesterol sequestering	-	[140]
Itraconazole	Oxysterol binding protein inhibitor	-	[141]
Monensin	Endosome acidification inhibitor	-	[142]
Niclosamide	Endosome acidification inhibitor	-	[143]
EIPA	Endosome acidification inhibitor	-	[144]
Bafilomycin A	Endosome acidification inhibitor	-	[145]
Ammonium Chloride	Endosome acidification inhibitor	-	[146]
Cytochalasin D (CytD)	Actin-polymerisation inhibitor	-	[145]
Budesonide	Autophagy activator	-	[147]
DDD85646	N-myristoyltransferase inhibitor	-	[124]
ML-9	Myosin light chain kinase inhibitor	-	[148]
sMBP-V33333	Receptor mimetic	-	[149]
sICAM-1	Receptor mimetic	-	[150,151,152]
Dendrimers	Blocking receptor attachment of EV-A71 and HIV	-	[24,153]
Deoxyglucose	Metabolism modifier	-	[154]
Azithromycin	Immunomodulatory	-	[155]
LL-37	Cathelicidins	-	[156]
PDS	Targeting vRNA	-	§
PhenDC3	Targeting vRNA	-	§

Abbreviations: nrp—no result posted; cwr—completed with result; w—withdrawn; * no publication related to *Enteroviruses* available. Clinical trial repositories: ClinicalTrial.gov, accessed on 1 September 2021 and ClinicalTrialsRegister.eu, accessed on 1 September 2021. § unreviewed manuscript preprint [157].

## Data Availability

The data presented in this study are integrally available here and in the following scientific articles (DOI:10.3390/v12070723 (accessed on 1 September 2021) and DOI:10.21203/rs.3.rs-646190/v2 (accessed on 1 September 2021)).

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
