# Peer review of "Rhinovirus Inhibitors: Including a New Target, the Viral RNA"

_viruses, 2021, doi:10.3390/v13091784_

Round 1

Reviewer 1 Report

This is an interesting review by respected authors which is mostly well written and introduces the very valid subject of the viral RNA as an antiviral target for the enteroviruses.  

My major concern is that the part of the review focusing on the RNA is dominated by the author’s own work, including four figures and descriptions of both published and unpublished work. For a review this feels a little unbalanced. It is perhaps the case that the authors are the major (or only) people working in this area but it still feels rather strange to write a review with so much focus on your own work. I would not feel comfortable publishing a review containing such emphasis on my own work.

The bias towards the authors own work continues to other areas. The concept of packaging signals has become quite well established in multiple picornavirus genera but is dismissed here as being ‘still debated’ with no further explanation given. Then the authors own paper on a related finding is instead described in more detail.  The table of inhibitors of cellular proteins also gives priority to the authors own papers while omitting those of others. Thus is disappointing.   

Some of the references are not correct or duplicated. I have not checked them all but would urge the authors to check.

Line 248 – 251 this section is confusing. At first I thought the reference to figure 1 was a mistake.  It would be useful to explain that the samples in fig 2 are the same (or similar) to those from fig 1.

Line 356-373 this section is confusing. Is PDS enhancing exit or impairing exit? It sounds like it is doing both!

Line 28 suggests that all enteroviruses are responsible for severe disease. I don’t think this is the case.

Line 183 better to say “…plant expression system.”

Author Response

Comments and Suggestions for Authors

This is an interesting review by respected authors which is mostly well written and introduces the very valid subject of the viral RNA as an antiviral target for the enteroviruses. 

>>> Thank you very much!

My major concern is that the part of the review focusing on the RNA is dominated by the author’s own work, including four figures and descriptions of both published and unpublished work. For a review this feels a little unbalanced. It is perhaps the case that the authors are the major (or only) people working in this area but it still feels rather strange to write a review with so much focus on your own work. I would not feel comfortable publishing a review containing such emphasis on my own work.

>>> We completely agree with this reviewer as we were aware of this lack of balance when asking the editors whether this was acceptable: “…. let me know whether its format (half review half report on results to be published elsewhere and therefore, not given in all details) appears acceptable to you.”. The answer:  “…. I don't see any problem to mix your results with the work of others. Maybe the structure can be rearranged a bit to briefly discuss the existing inhibitors detailed in the table and possibly add those targeting RNA.”

>>> Maybe we should not call it a ‘review’ but something else – we would be happy with a suggestion!

The bias towards the authors own work continues to other areas. The concept of packaging signals has become quite well established in multiple picornavirus genera but is dismissed here as being ‘still debated’ with no further explanation given. Then the authors own paper on a related finding is instead described in more detail.  The table of inhibitors of cellular proteins also gives priority to the authors own papers while omitting those of others. Thus is disappointing.

>>> We are thankful for drawing our attention to this lack of space and credit given to the seminal work of Peter Stockley and others who discovered stretches of RNA sequences within the genomes of parecho- and enterovirus with propensity to bind the inner face of the protein shell and thereby acting as multiple degenerate packaging signals. To mitigate this deficiency we substantially expanded the discussion of the concept of packaging signals in PARECHO- and ENTEROVIRUSES and their discovery via RNA SELEX and other methodology. We have also added a visual and extended the tables to overcome this imbalance towards our own work.

Some of the references are not correct or duplicated. I have not checked them all but would urge the authors to check.

>>> We checked, counter checked, and corrected wrong references

 Line 248 – 251 this section is confusing. At first I thought the reference to figure 1 was a mistake.  It would be useful to explain that the samples in fig 2 are the same (or similar) to those from fig 1.

 >>> Several sentences related to these figures were completely re-written. We also added a new Figure (Fig. 1, a visual) indicating the various drug targets along the viral replication cycle. Thus, the figure numbers have changed. We have taken care of explaining that the samples in the old Figs 1 and 2 were prepared identically. See Legend to new Fig. 3.

Line 356-373 this section is confusing. Is PDS enhancing exit or impairing exit? It sounds like it is doing both!

>>> We now made it clear that incubation of RV-A2 with PDS under conditions allowing for its diffusion through the proteinaceous capsid (34 °C) makes the RNA accessible for Syto 82 already at lower temperature as compared to the control incubation (at 4°C where diffusion of PDS into the virion is minimal). However, counterintuitively, in vitro the uncoating temperature (i.e. RNA exit) is essentially the same. Nevertheless and most importantly, virus treated with PDS at 34°C uncoats less efficiently in vivo

Line 28 suggests that all enteroviruses are responsible for severe disease. I don’t think this is the case.

>>> We modified the text to clarify the undebated fact that not all enteroviruses are responsible for severe disease: We explicitly mention the most dangerous ones, poliovirus, enterovirus-71, and enterovirus-D68.

Line 183 better to say “…plant expression system.

>>> done

Reviewer 2 Report

Authors of the manuscript entitled “Rhinovirus Inhibitors: Including a new Target, the Viral RNA” discuss an interesting idea of targeting rhinovirus genomic RNA by small molecules that can potentially inhibit viral infection. The original idea comes from the drugs in development for cancer therapy targeting nucleic acid structures called G-quadruplexes (GQs). GQs have been described in human genome, but they also have been found in viral genomes including RNA viruses (for example, Ruggiero and Richter, NAR 2018). As it is, the manuscript is, however, skewed towards the viral RNA (vRNA) as a potential drug target. The review would make more sense if authors would discuss about all other types of rhinovirus inhibitors with pros and cons for each group more broadly; how these inhibitors work/do not work for other picornaviruses and/or other viruses. It would be useful to add relevant information on vRNA used as a target in other viruses, for example in HIV-1 (Perrone et al, J Antimicrob Chemother 2014). This needs re-structuring of the review and adding missing information. The review has several figures but they all present unpublished data rather than summarizing schematics. I suggest thinking on the main idea of the manuscript and adding a visual to the manuscript. Unpublished manuscripts cited here have to be accepted before this review is published. Some specific comments follow below.

Title does not reflect the content of the review. Title promises to present on rhinovirus inhibitors, but text is mainly on picornaviral RNA biology. Authors should consider changing the title to reflect the text or alternatively expand the text on rhinovirus inhibitors working on other than vRNA targets. RNA biology is very interesting, but the focus should be more on the viral infectious cycle inhibitors and their mode of action, if authors review on targets and inhibitors. Inhibitors targeting non-vRNA should be discussed also in the text, not only listed in the tables (Table 2 and Table 3). Part of the discussion is based on the unpublished manuscript(s) (lines 311, 356, 384); manuscript(s) have to be accepted before publication of this review to ensure proper referencing.

GQs are present in human genome, so targeting GQs will potentially lead to side effects, which are not discussed in this review manuscript. The GQs-targeting drug selectivity issue should be brought up.

Figures 1 and 2. It is not clear if data in Figs 1 and 2 have been published before or data are produced for this review.

Table 1 caption should be ‘List of capsid binders’. Compound 17 does not target the hydrophobic pocket. It’s main target is an inter-protomer pocket composed of amino acids from two VP1 and one VP3 chains. CP17 does increase capsid stability, though.

Section 3 original refs missing. Looks incomplete, just brief mentioning of inhibitors, it also has lines 139-146 which do not belong to ‘Targeting non-structural viral proteins’ as it talks about viral RNA.

152 – 154 references missing

Line 267 Sentence ‘… although its overall shape is conserved even in the absence of the protein capsid’. Word ‘conserved’ is somewhat misleading; at this resolution the ‘overall shape’ of RNA is a ball (RNA can have different secondary/tertiary structure there). I suggest using other word like ‘retained’ or ‘preserved’.

Line 283 Following sentence is unclear. ‘So far, we only considered double-stranded RNA stretches interrupted by single-stranded regions, usually appearing as loops in folding predictions [100].’ Do authors mind considered in this manuscript?

Line 293 Please add the reference for ‘Such GQ stabilisers are currently under focus for their potential use as anticancer drugs.’

Abbreviations should be introduced when first mentioning term in the text, eg pyridostatin (PDS) is mentioned first in line 292 but abbreviation appears in line 314.

Line 296 please provide the reference for ‘Harutyunyan and colleagues assumed that RNA exit from the shell requires at least some unfolding.’

Line 357 phosphate buffered saline (PBS)

Table 3 what sGAGs stands for? The capture for table 3 is not appropriate as PDS for example still targets the virion or vRNA, not cellular proteins.

Line 335 Sentence is not clear needs rephrasing: ‘Song et al. [145] showed that massive alteration of the poliovirus P1 RNA sequence (i.e. encoding the capsid proteins) produces a variant with infectivity comparable to that of native virus provided that the encoded amino acid sequence and the GC content were maintained [146]’

Line 361 ‘Molecular diffusion of is very low at 4 °C.’ Molecular diffusion of what?

ref 131 in line 363 has nothing to do with PDS.

Line 373 ref 151 does not describe ‘Pre-incubation of RV-A2 with PDS followed by challenge of HeLa cells …’

Section in lines 390-401 has no references at all, please add.

Line 392 Please add reference for this observation: ‘In agreement with the strikingly different effect of sodium and potassium ions at the RNA level …’

Line 431 Please add reference for this: ‘Furthermore, although PDS and Phen-DC3 only weakly inhibited RV infection,’

Considering tables, one useful piece of information would be chemical structures of compounds.

Figure 3. Why there are two images in panel A? What is the difference between the image on right hand side vs the image on left hand side in Fig 3A?

The layout of the images in the figures could be unified.

Author Response

Comments and Suggestions for Authors

Authors of the manuscript entitled “Rhinovirus Inhibitors: Including a new Target, the Viral RNA” discuss an interesting idea of targeting rhinovirus genomic RNA by small molecules that can potentially inhibit viral infection. The original idea comes from the drugs in development for cancer therapy targeting nucleic acid structures called G-quadruplexes (GQs). GQs have been described in human genome, but they also have been found in viral genomes including RNA viruses (for example, Ruggiero and Richter, NAR 2018). As it is, the manuscript is, however, skewed towards the viral RNA (vRNA) as a potential drug target. The review would make more sense if authors would discuss about all other types of rhinovirus inhibitors with pros and cons for each group more broadly; how these inhibitors work/do not work for other picornaviruses and/or other viruses. It would be useful to add relevant information on vRNA used as a target in other viruses, for example in HIV-1 (Perrone et al, J Antimicrob Chemother 2014). This needs re-structuring of the review and adding missing information. The review has several figures but they all present unpublished data rather than summarizing schematics. I suggest thinking on the main idea of the manuscript and adding a visual to the manuscript. Unpublished manuscripts cited here have to be accepted before this review is published. Some specific comments follow below.

>>> We fully agree with this reviewer in that a discussion of the pros and cons of the other inhibitors might be useful in a dedicated review. However, we rather wanted to shortly discuss, as a kind of Introduction, the so far known viral and cellular targets and their potential inhibitors to better illustrate that the RNA has so far not been considered a potential drug target at all, at least in ENTEROVIRUSES. We corresponded with the editors prior to submission and asked whether such a format was acceptable and they agreed. In the now modified version we refer to the excellent review on potential drug targets in ENTEROVIRUSES [van der Linden, L., Wolthers, K.C., and van Kuppeveld, F.J. (2015). Replication and Inhibitors of Enteroviruses and Parechoviruses. Viruses 7, 4529-4562], to justify why we only shortly discuss them. We now put the emphasis on more recent discoveries (i.e. after 2015, the date of the above review). We also followed the suggestion of this reviewer to very shortly discuss the effect and mode of action of G-quadruplex stability modifiers on the so far studied viruses. For example, in ‘Outlook’, we mention that GQ-stabilizers have only been investigated with respect to modification of translation/transcription of viral nucleic acids and NOT in the context of genome uncoating. We now also cite (Perrone et al, J Antimicrob Chemother 2014) and other workers who have shown the effect of GQ stabilizers on replication of viruses.

Title does not reflect the content of the review. Title promises to present on rhinovirus inhibitors, but text is mainly on picornaviral RNA biology. Authors should consider changing the title to reflect the text or alternatively expand the text on rhinovirus inhibitors working on other than vRNA targets. RNA biology is very interesting, but the focus should be more on the viral infectious cycle inhibitors and their mode of action, if authors review on targets and inhibitors. Inhibitors targeting non-vRNA should be discussed also in the text, not only listed in the tables (Table 2 and Table 3). Part of the discussion is based on the unpublished manuscript(s) (lines 311, 356, 384); manuscript(s) have to be accepted before publication of this review to ensure proper referencing.

>>> We are convinced that the title is clear and unmistakably indicates that we here concentrate on the RNA as a novel target for viral inhibitors (see also above). This is made even clearer in the Abstract that explicitly indicates that we are focusing onto the RNA and much less onto protein targets that are only being mentioned briefly to indicate that the RNA has not been considered at all as a target. Nevertheless, we have now made a number of adjustments, in particular, adding more recently identified inhibitors, i.e. dendrimers, nanoparticles, and filament-forming compounds, as well as cellular proteins necessary for viral propagation. Our short review should thus be rather considered an update of the one of 2015 mentioned above. A copy of the non-reviewed manuscript is now available and referenced at the appropriate locations as ‘unreviewed manuscript DOI: 10.21203/rs.3.rs-646190/v2’. We agree in that this manuscript is not exactly a ‘Review’ and corresponded with the editors about the format. They told us that this format was acceptable. We shall be happy with using another, more appropriate description for the text.

GQs are present in human genome, so targeting GQs will potentially lead to side effects, which are not discussed in this review manuscript. The GQs-targeting drug selectivity issue should be brought up.

>>> This is a very good point! We picked it up and expanded the text accordingly to mention this.

Figures 1 and 2. It is not clear if data in Figs 1 and 2 have been published before or data are produced for this review.

>>> These figures (now Fig. 2 and 3) are new and unpublished. This is now clarified in the text.

Table 1 caption should be ‘List of capsid binders’. Compound 17 does not target the hydrophobic pocket. It’s main target is an inter-protomer pocket composed of amino acids from two VP1 and one VP3 chains. CP17 does increase capsid stability, though

>>> We corrected this by adding a foot note. We also mention that compound 17 also increases capsid stability, although by ‘clamping together’ protomers and not by inhibiting movements within the pocket usually containing the pocket factor.

Section 3 original refs missing. Looks incomplete, just brief mentioning of inhibitors, it also has lines 139-146 which do not belong to ‘Targeting non-structural viral proteins’ as it talks about viral RNA.

>>> modified accordingly

152 – 154 references missing

>>> references added

Line 267 Sentence ‘… although its overall shape is conserved even in the absence of the protein capsid’. Word ‘conserved’ is somewhat misleading; at this resolution the ‘overall shape’ of RNA is a ball (RNA can have different secondary/tertiary structure there). I suggest using other word like ‘retained’ or ‘preserved’.

>>> rephrased

Line 283 Following sentence is unclear. ‘So far, we only considered double-stranded RNA stretches interrupted by single-stranded regions, usually appearing as loops in folding predictions [100].’ Do authors mind considered in this manuscript?

>>> The sentence was modified to make it clear that we are referring to our present manuscript.

Line 293 Please add the reference for ‘Such GQ stabilisers are currently under focus for their potential use as anticancer drugs.’

>>> done and we now state, in ‘Outlook’ that a quick Google search for ‘G-quadruplex’ yields more than 1.8 mio hits.

Abbreviations should be introduced when first mentioning term in the text, eg pyridostatin (PDS) is mentioned first in line 292 but abbreviation appears in line 314.

>>> done

Line 296 please provide the reference for ‘Harutyunyan and colleagues assumed that RNA exit from the shell requires at least some unfolding.’

>>> done

Line 357 phosphate buffered saline (PBS)

>>> done

Table 3 what sGAGs stands for? The capture for table 3 is not appropriate as PDS for example still targets the virion or vRNA, not cellular proteins.

>>> corrected, ‘glycoseaminoglycans’ is now written out

Line 335 Sentence is not clear needs rephrasing: ‘Song et al. [145] showed that massive alteration of the poliovirus P1 RNA sequence (i.e. encoding the capsid proteins) produces a variant with infectivity comparable to that of native virus provided that the encoded amino acid sequence and the GC content were maintained [146]’

>>> was entirely rephrased

Line 361 ‘Molecular diffusion of is very low at 4 °C.’ Molecular diffusion of what?

>>> we are referring here to the diffusion of small molecules through the proteinaceous capsid wall. The sentence was rephrased.

ref 131 in line 363 has nothing to do with PDS.

>>> corrected

Line 373 ref 151 does not describe ‘Pre-incubation of RV-A2 with PDS followed by challenge of HeLa cells …’

>>> corrected

Section in lines 390-401 has no references at all, please add.

>>> done

Line 392 Please add reference for this observation: ‘In agreement with the strikingly different effect of sodium and potassium ions at the RNA level …’

>>> refers to the data shown in Figs. 1 and 2 (now Fig. 2 and 3) and the unreviewed manuscript. This is now explicitly specified and cited

Line 431 Please add reference for this: ‘Furthermore, although PDS and Phen-DC3 only weakly inhibited RV infection,’

>>> reference to the unreviewed manuscript added

Considering tables, one useful piece of information would be chemical structures of compounds.

>>> chemical structures added as Fig. 4

Figure 3. Why there are two images in panel A? What is the difference between the image on right hand side vs the image on left hand side in Fig 3A?

>>> These are two different micrographs of identically prepared samples. In the left one the filaments look somewhat like pearls-on-a-string. This is now mentioned in the legend.

The layout of the images in the figures could be unified.

>>> where possible (e.g. in the unpublished figures) this was done

Many parts of the manuscript have been modified based on all three reviewers’ comments and all references were carefully checked

Reviewer 3 Report

Following, I do some comments/suggestions that authors should incorporate.
1. Although targeting cellular proteins undoubtedly avoid escape mutants appearance, on can expect unwanted side effects. Can the authors discuss about specific drawbacks of interfering cellular targets?
2. By one side, it is stated that viral RNA is not under pressure (line 209), but latter it is stated that  “the tightly folded RNA has a crystalline-like consistency” (line 247). Please, clarify this.
3. From many previous works, it is clear the RNA has any role during encapsidation and uncoating. Perhaps the authors should cite recent results about protein capsid RNA interaction and rearrangements during uncoating (Hrebík, D. et al. ICAM-1 induced rearrangements of capsid and genome prime rhinovirus 14 for activation and uncoating. PNAS 2021, 118 (19), e2024251118).
4. Line 51: the link does not work.
5. Line 84: expand the discussion on the newly discovered inhibitory site present in the capsid (refs. 27 and 28), specially in regard on its mechanism of action.

Author Response

Comments and Suggestions for Authors

Following, I do some comments/suggestions that authors should incorporate.

  1. Although targeting cellular proteins undoubtedly avoid escape mutants appearance, on can expect unwanted side effects. Can the authors discuss about specific drawbacks of interfering cellular targets?

>>> where known, e.g. from knockout of the potential cellular target (e.g. experiments with pla2g16 knockout in mice) we have added this information.

  1. By one side, it is stated that viral RNA is not under pressure (line 209), but latter it is stated that “the tightly folded RNA has a crystalline-like consistency” (line 247). Please, clarify this.

>>> we wanted to explain that the RNA was not exerting any pressure on the capsid upon confinement, i.e. it would not expand when the capsid is removed; this is not in contradiction to a “crystalline-like consistency”. It is very much different from the dsDNA pumped into a phage head; in this case, the nucleic acid is confined at pressures of several tens of atmospheres. This is now made clearer by re-phrasing the relevant sentences.

  1. From many previous works, it is clear the RNA has any role during encapsidation and uncoating. Perhaps the authors should cite recent results about protein capsid RNA interaction and rearrangements during uncoating (Hrebík, D. et al. ICAM-1 induced rearrangements of capsid and genome prime rhinovirus 14 for activation and uncoating. PNAS 2021, 118 (19), e2024251118).

>>> at the time of finishing this manuscript, the above paper was not yet available to us. We now cite it in the following part of the text when the energy is discussed:

“At least part of it might stem from interactions with the receptor (ICAM-1) at low pH as observed for RV-B14; it leads to a rearrangement of the RNA around the two-fold axis involving the loss of interaction with TRP 2038 [179]; this is in accordance with nanoDSF measurements of TRP fluorescence changes during in vitro uncoating of RV-A2 via heating [180].”

  1. Line 51: the link does not work.

>>> this must have to do with the conversion into a pdf file. It works in the original word file.

  1. Line 84: expand the discussion on the newly discovered inhibitory site present in the capsid (refs. 27 and 28), specially in regard on its mechanism of action.

>>> explained and mechanism of action shortly mentioned.